# *Populus**PtERF85* Balances Xylem Cell Expansion and Secondary Cell Wall Formation in Hybrid Aspen

**DOI:** 10.3390/cells10081971

**Published:** 2021-08-03

**Authors:** Carolin Seyfferth, Bernard A. Wessels, Jorma Vahala, Jaakko Kangasjärvi, Nicolas Delhomme, Torgeir R. Hvidsten, Hannele Tuominen, Judith Lundberg-Felten

**Affiliations:** 1Umeå Plant Science Centre, Department of Plant Physiology, Umeå University, SE-90187 Umeå, Sweden; carolin.seyfferth@psb.vib-ugent.be (C.S.); bernard.wessels@umu.se (B.A.W.); torgeir.r.hvidsten@nmbu.no (T.R.H.); 2Organismal and Evolutionary Biology Research Programme, Faculty of Biological and Environmental Sciences, University of Helsinki, FI-00014 Helsinki, Finland; vahalajorma@gmail.com (J.V.); jaakko.kangasjarvi@helsinki.fi (J.K.); 3Umeå Plant Science Centre, Department of Forest Genetics and Plant Physiology, Swedish University of Agricultural Sciences, SE-90184 Umeå, Sweden; nicolas.delhomme@slu.se (N.D.); Hannele.Tuominen@slu.se (H.T.); 4Faculty of Chemistry, Biotechnology and Food Science, Norwegian University of Life Sciences, 1433 Ås, Norway

**Keywords:** cell wall thickness, ERF85 (CRF4), lignin, ribosome biogenesis, wood development, xylem expansion

## Abstract

Secondary growth relies on precise and specialized transcriptional networks that determine cell division, differentiation, and maturation of xylem cells. We identified a novel role for the ethylene-induced *Populus Ethylene Response Factor PtERF85* (*Potri.015G023200*) in balancing xylem cell expansion and secondary cell wall (SCW) formation in hybrid aspen (*Populus tremula x tremuloides*). Expression of *PtERF85* is high in phloem and cambium cells and during the expansion of xylem cells, while it is low in maturing xylem tissue. Extending *PtERF85* expression into SCW forming zones of woody tissues through ectopic expression reduced wood density and SCW thickness of xylem fibers but increased fiber diameter. Xylem transcriptomes from the transgenic trees revealed transcriptional induction of genes involved in cell expansion, translation, and growth. The expression of genes associated with plant vascular development and the biosynthesis of SCW chemical components such as xylan and lignin, was down-regulated in the transgenic trees. Our results suggest that PtERF85 activates genes related to xylem cell expansion, while preventing transcriptional activation of genes related to SCW formation. The importance of precise spatial expression of *PtERF85* during wood development together with the observed phenotypes in response to ectopic *PtERF85* expression suggests that PtERF85 contributes to the transition of fiber cells from elongation to secondary cell wall deposition.

## 1. Introduction

Secondary xylem (wood) cells undergo cell expansion, extensive secondary cell wall (SCW) thickening, and ultimately programmed cell death. SCW formation involves the deposition of cellulose, xylan, and lignin (reviewed in [1,2]). Understanding the molecular regulation of SCW formation in xylem vessels and fibers has become of special interest in tree research for breeding approaches and for generating a toolbox to tailor wood biomass production. High-resolution transcriptomic analysis of cambial tissues and developing xylem of aspen (*Populus tremula*) stems revealed transcriptional networks and putative molecular regulators of the different phases of xylem differentiation [3] that can be targeted for reverse genetics. Functional analyses have furthermore revealed a set of WRKY, NAC, and MYB transcription factors (TFs) as regulators of SCW biosynthesis [4,5,6,7,8,9,10,11,12].

The plant hormone ethylene has recently been shown to impact SCW formation in woody tissues. Gravitropic and exogenous stimulation of ethylene signaling leads to enhanced cambial activity, increased fiber-to-vessel ratio, and to the formation of cellulose-rich gelatinous layers (G-layers) in xylem fibers, while these responses were abolished in transgenic ethylene-insensitive hybrid aspen trees [13,14]. Ethylene-regulated TFs from the PtEIN3 (Ethylene Insensitive) and the PtERF (Ethylene Response Factor) family are potential transcriptional regulators of SCW development in aspen trees [15,16,17,18]. Computational network analysis predicted PtEIN3D and eleven PtERFs as hub TFs in transcriptional regulation during stem development in *P. tremula* [17]. These *TFs* are co-expressed with genes related to lignan and xylan biosynthesis [17]. Furthermore, results from transgenic hybrid aspen (*P. tremula x tremuloides*) trees that overexpress *PtERF139*, or a dominant negative version of this *TF*, suggest a regulatory function of PtERF139 for SCW deposition and lignin biosynthesis [18]. In addition, ectopic expression of four other *PtERFs* affected stem diameter and wood biochemistry in hybrid aspen trees, indicating a functional link between *PtERFs* and wood formation [15]. One of the *PtERFs* identified in the aforementioned study was *PtERF85*. Overexpression of *PtERF85* under the wood-specific (*pLMX5*) promoter reduced hybrid aspen stem diameter and height growth and altered the chemical wood composition as identified in a large-scale screening of *PtERFs* with an impact on wood formation [15]. In the present article, we explore the phenotype caused by *PtERF85* overexpression to further understand the function of PtERF85 in xylem formation. Expression of *PtERF85* is enhanced in hybrid aspen stems after ethylene treatment [15], but not in ethylene-insensitive hybrid aspen stems [14], suggesting that its function is controlled by ethylene. The closest homolog of *PtERF85* in *Arabidopsis thaliana*, *AtCRF4 (AT4G27950)*, belongs to an evolutionarily distinct subgroup of *ERFs* called Cytokinin Response Factors (*CRFs*; [19]). All *CRFs*, except *AtCRF4*, can be induced by exogenous cytokinin application [20]. The nine members of this subgroup can form homo- and heterodimers [21], suggesting that they might function together in gene regulation. Indeed, knock-out or overexpression of *AtCRF4* alone in Arabidopsis resulted in a wildtype (WT) like phenotype under normal growth conditions, further supporting the idea of partial functional redundancy among the CRFs [22,23]. On the contrary, ectopic expression of *PtERF85* alone was sufficient to modify tree growth and xylem cell wall chemistry [15].

In this study, we show how ectopic expression of *PtERF85* alters xylem formation by decrypting the biological processes and gene targets downstream of PtERF85 in xylem tissues. For this, a detailed phenotypic characterization of the transgenic *PtERF85* overexpressing trees (ERF85OE) was combined with a large-scale xylem transcriptome analysis using RNA-sequencing.

## 2. Materials and Methods

### 2.1. Plant Growth Conditions

Hybrid aspen (*Populus tremula L. x P. tremuloides* Michx, clone T89) was used in all experiments (from here on defined as wildtype (WT)), propagated in vitro, and transferred to soil. Greenhouse growth conditions were as follows; 18:6 h day:night cycle, 20:15 °C day:night temperature, and relative humidity ranging between 50 and 60%. All trees were grown in a commercially available sand/soil/fertilizer mix (Krukjord, Hasselfors Garden, Örebro, Sweden) and fertilized once per week with approximately 150 mL 1% Rika-S (N/P/K 7:1:5; Weibulls Horto, Hammenhög, Sweden) starting the third week after transplanting and ending one week before harvest. Trees were rotated weekly to minimize positional effects. Growth conditions for trees used for wood density and anatomy analysis were slightly different, as they were performed in a different laboratory (Helsinki University, Helsinki, Finland), yet similar height growth phenotypes were observed as in the other experiments: trees were grown under greenhouse conditions in a mixture of peat, sand, and Vermiculite (6:2:1, *v/v/v*), and fertilized with 2.5 g/L (*w/v*) of Osmocote Exact Hi-Start (N/P/K 15:4:8; Scotts). Growth conditions were as follows; 18:6 h day:night cycle, and 20:18 °C day:night temperature. Natural daylight intensity (on average 400 µmol/ms) was controlled with curtains and supplemented with high-pressure sodium lamps when needed.

### 2.2. Cloning and Generation of Transgenic Trees

*pLMX5::PtERF85* overexpressor constructs and plant transformation have been previously described [15]. To generate the *pPtERF85::GUS* construct, a 1951-bp promoter region upstream of *PtERF85* (*Potri.015G023200*) was amplified from *Populus trichocarpa* genomic DNA. The primers used can be found in Appendix A. The PCR product was cloned into the pDONR221 donor vector and recombined to pKGWFS7, driving eGFP and GUS expression [24], thereafter cloned into the Agrobacterium strain GV3101 pMP90 and transformed into *Populus tremula x Populus tremuloides* [25]. At least 17 independent transgenic lines carrying the *pPtERF85::GUS* construct were generated and selected following the detection of GUS expression in stems, leaves, and roots of in vitro grown trees.

### 2.3. Histochemical Staining

GUS staining in wood material from greenhouse-grown transgenic hybrid aspen trees expressing the *pPtERF85::GUS* construct was performed in three independent transgenic lines. Trees were grown in the greenhouse to a height of approximately 2 m. Stem segments from 10 cm above soil level were used for histochemical staining. For the promoter activity analysis in response to ACC (1-aminocyclopropane-1-carboxylic acid), the transgenic lines were grown in an in vitro culture system for four weeks. A ten-hour treatment of either 100 µM ACC or mock (water) was applied (according to [14]). Three stem pieces from the fifth internode down were harvested and the sections shown are representing a region in the upper end of the lower third of the stem. Plant material was fixed in 90% acetone for 20 min at room temperature. Acetone was dried off and samples were immersed in the GUS-staining solution containing 1 mM X-GlcA (5-bromo-4-chloro-3-idolyl glucuronide), 50 mM K-phosphate buffer (pH 7.0), 0.1% Triton, 1 mM potassium ferricyanide (K_3_Fe(CN)_6_), and 1 mM potassium ferrocyanide (K_4_Fe(CN)_6_). Samples were vacuum infiltrated for 3 cycles of 2 mins, followed by incubation at 37 °C in the dark and terminated after 6 h (for greenhouse-grown trees) or 16 h (for ACC-treated in vitro-grown trees). Samples were rinsed with phosphate buffer, cleared using an ethanol washing series, and incubated in 100% ethanol overnight. Afterwards, an ethanol washing series was used to rehydrate the samples. GUS expression was analyzed in 60–70 µM thick stem cross-sections (cut using a vibratome) that were mounted in 50% glycerol and imaged with a Zeiss Axioplan2 microscope, Axiocam HRc camera and Axiovision V4.8.2 software (Carl Zeiss Light Microscopy, Göttingen, Germany). To check the start of SCW formation, stem cross-sections (50 µm) were made using the vibratome and stained with Safranin:Alcian blue (1:2 ratio) for 1 min. Sections were rinsed twice in water and mounted in 50% glycerol.

### 2.4. Wood Density Measurements

A 20 cm piece of stem (*n* = 5) from a (approximately) 2 m tall tree was cut 10 cm above the soil and oven-dried for one week at 65 °C. Wood density was determined as the dry weight of the wood sample divided by the dry volume. The diameter of each stem sample was measured (with a caliper) from different positions (top, middle, and bottom of the stem piece) three times. Each stem piece was measured independently three times (as technical replicates) to ensure the correct volume.

### 2.5. Transmission Electron Microscopy

Approximately 0.5 mm thick transverse hand-cut sections from greenhouse-grown stem pieces harvested 8–10 cm above the soil level were fixed in 2.5% glutaraldehyde in sodium cacodylate buffer (0.1 M, pH 7.2) for more than 48 h, treated for 2 h with 1% OsO4 and then processed, embedded, sectioned, and imaged as described in [26].

### 2.6. Quantification of Xylem Cells, Fiber Diameter and Cell Wall Thickness

Ultrathin stem sections (1 µm thick) were stained with toluidine blue and pictures were taken with a 20-fold and 100-fold magnifying objective of a Zeiss Axioplan 2 microscope at approximately 1 mm inwards from the cambium. Each picture covers an area of approximately 3.02 mm^2^ or 149,295 µm^2^, respectively. The number of vessels and ray cells were counted in ImageJ using a picture taken with a 20-fold magnification. The same pictures were used to quantify vessel diameter, defined as the area surrounded by the vessel cell wall. Pictures taken with a 100-fold magnifying objective were used to quantify fiber number, diameter, and lumen to lumen distance (from here on defined as SCW thickness). Sections from three trees per genotype were analyzed and fiber parameters were quantified on five different positions per section (within an area located approximately 1 mm inwards from the cambium). The dark-blue-stained middle lamella was used to define the boundary of each fiber cell, and the area included by it was defined as the fiber cell diameter. Cell wall thickness, defined as the distance between the lumen of two neighboring cells, was measured at 50 randomly chosen spots per picture. To calculate statistical differences between fiber numbers, diameter, and cell wall thickness in WT and the independent transgenic lines, measurements from all five pictures taken from each tree were summarized to calculate the mean value.

### 2.7. RNA Extraction

RNA was extracted from 15 cm long stem pieces taken approximately 15–30 cm above the soil. Prior to RNA extraction, bark and pith of each stem piece were removed and the developing and mature xylem (cells with high expression of *PtERF85* in ERF85OE) was scraped all around the stem and ground in liquid nitrogen with a mortar and pestle. RNA was extracted from three WT samples (pooled from three trees each) and three samples (each a pool of three trees) for each line of the ERF85OE. RNA was used for RNA-Sequencing (RNA-Seq) and real-time quantitative PCR (qPCR). RNA extraction was performed using the BioRad Aurum RNA extraction kit (Hercules, CA, United States) following the manufacturer’s instructions, except for the on-column DNase treatment which was replaced by a DNase treatment using DNAfreeTM (Ambion, Waltham, MA, USA) after RNA elution from the column. RNA was cleaned using the Qiagen MinElute kit (Hilden, Germany). RNA was quantified using a Nanodrop ND-1000 (Nano-Drop Technologies, Wilmington, DE, USA) and the quality was assessed with an Agilent 2100 Bioanalyzer (Agilent, Waldbronn, Germany) with Agilent RNA 6000 Nano Chips according to the manufacturer’s instructions.

### 2.8. RNA-Seq Sample Preparation and Statistical Analysis

Sequencing library generation and paired-end (2 × 100 bp) sequencing using Illumina HiSeq 2000 were carried out at the SciLifeLab (Science for Life Laboratory, Stockholm, Sweden). Raw files can be downloaded from the European Nucleotide Archive (ENA) under PRJEB35743. Data quality summary and scripts used in this study are available in the GitHub repository https://github.com/carSeyff/PtERF85OE v1.0.1 (accessed on 18 May 2021) (doi:10.5281/zenodo.4770645), where details on non-default parameters for data pre-processing can be found. Briefly, data pre-processing was performed as described in Delhomme et al. (2014) and consisted of removal of ribosomal RNAs and sequencing adapters (using SortMeRNA (v2.1b using the rRNA libraries shipped with the tool) and Trimmomatic (v0.32) [27,28]), as well as quality-based trimming by the latter. The *P. trichocarpa* (Pt) genome (version 3.0) was retrieved from http://popgenie.org (accessed on 1 March 2018) and used for the alignment of the quality-filtered and trimmed read pairs using STAR (v2.4.2a). Read counting per gene and library was performed using HTSeq that discard ambiguous and multi-mapping reads [29]. Statistical data analysis was done in R (version 3.4.2) using DESeq2 (version 1.22.2 [30]). This analysis included library size adjustment to obtain the size factor value for each genotype (WT against ERF85OE lines) and variance stabilizing transformation (VST). The resulting normalized read counts were used for principal component analysis. The differential gene expression analysis between the WT and the three lines of the ERF85OE was conducted on the raw counts. To identify PtERF85-mediated transcriptional changes and independent on-line-to-line variation, data from all three ERF85OE lines were combined for further analysis. Similarly, we also combined all three WT replicates. Principal component analysis (PCA) was used to determine the variance between the obtained transcriptomes and was performed according to the standard setting in the DESeq2 package. Differentially expressed genes (DEGs) were selected for ERF85OE compared to WT and defined by a |log2FC| > 1 and a pAdj < 0.01 (given in Appendix A). Heatmaps were generated in R with the pheatmap package (version 1.0.12) and gplots. All given *Populus* gene annotations are reported according to the gene name given to its closest Arabidopsis homolog.

### 2.9. Validation of RNA-Seq Results by qPCR

RNA used to validate gene expression changes in ERF85OE lines was extracted from three replicates per genotype and three pools of WT trees. cDNA was synthesized from 1 µg RNA using iScript cDNA Synthesis Kit (BioRad, Hercules, CA, USA). Each qPCR reaction (in total 15 µL) contained 2 µL of diluted cDNA (1:4), 7.5 µL 2xSYBR Green Mastermix (Roche, Basel, Switzerland) and 2.75 µL of 10 nM forward and reverse primers (listed in Appendix A) and run in the Bio-Rad CFX96 Real-Time System (Hercules, CA, USA). Target gene expression was compared with the expression of three reference genes (*PtACT1* and *PtUBQ-L*), which showed stable expression patterns in poplar stems (Wang et al., 2016). *PtUBQ-L* was chosen as the reference gene for calculating gene expression changes (presented as log2 of ΔcT).

### 2.10. Bioinformatic and Statistical Analysis

Gene ontology and PFAM enrichment were performed using *P. trichocarpa* homologs in AspWood based on Blast2Go [3]. Statistical analysis of the phenotypic data was performed in R using one-way ANOVA and post-hoc HSD Tukey tests. Each experiment was replicated using at least three trees per transgenic line. Multiple comparison tests were done using the R package multcompView (version 0.1-7), using a cut-off of *p*-value < 0.05. Promoter motif analysis was done as described in [14]. In brief, promoter regions with different lengths (500 bp, 1 kbp, 1.5 kbp, and 2 kbp) were obtained from the *P. trichocarpa* genome using the webtool “sequence search” from Popgenie (http://popgenie.org/sequence_search, accessed on 16 November 2018). We were able to obtain promoter regions of 401 (down-regulated) and 1837 (up-regulated) genes and checked the occurrence of the GCC-Box (“AGCCGCC”), the ERF-Box (“GCCGCC”) or the DRE-box (“RCCGAD”) on both strands. The hypergeometric distribution was used to determine statistical enrichment of either of the motifs among down- or up-regulated genes.

## 3. Results

### 3.1. PtERF85 Is Expressed in the Cambium and Expanding Xylem

The *P. trichocarpa* (Pt) genome encodes 170 *ERF*s, subdivided into ten different clusters [15]. The *Cytokinin Response Factors (CRFs)* form a cluster in the ERF gene family (cluster VI; *PtERF81-PtERF88*) that comprises homologs to regulators of the root, shoot, leaf, and flower development identified in Arabidopsis [19,23,31]. Phylogenetic analyses identified eight *CRFs* in *P. trichocarpa* ([15]; Figure 1a). A high-resolution expression atlas derived from longitudinal sections through aspen stems [3], from here on referred to as the “AspWood” database, indicated high transcript abundances for all *PtCRF4* family members in samples representing cambial cell proliferation and xylem cell expansion (Figure 1a,b and Appendix A). Transcript levels of *PtERF85* were high in the phloem and vascular cambium, peaked in the xylem expansion zone, and dropped towards xylem maturation and secondary cell wall (SCW) formation before slightly increasing in the cell death zone. As ethylene is known to interfere with cambial activity and cell expansion [14], and high *PtCRF* expression was observed in the corresponding developmental zones, we explored the responsiveness of *PtCRFs* towards the ethylene precursor ACC in stems of WT trees and two ethylene-insensitive trees in published datasets (Figure 1c; [14]). Treatment with the ethylene precursor ACC for 10 h triggered induction of *PtERF81*, *PtERF82,* and *PtERF85* expression, and down-regulated the expression of *PtERF86* (Figure 1c) in stems of WT but not the ethylene-insensitive trees, suggesting that these *ERFs* operate downstream of the ethylene signaling pathway. Phylogenetic analysis indicated 80% sequence identity between *PtERF85* and *PtERF86* [15]. The high sequence similarity is typical for a gene duplication event and can be associated with redundant protein function. However, although *PtERF85* and *PtERF86* showed high gene and protein sequence similarity, and probably evolved through a whole-genome duplication event, their opposite transcriptional regulation in response to ACC suggests that their ethylene response and their potential function during wood development might be different from each other. Therefore, we assumed that *PtERF85* and *PtERF86* are not redundant and focused our further analysis only on the role of *PtERF85* in ethylene-regulated wood development. We examined the expression of *PtERF85* in stem tissues of transgenic hybrid aspen (*P. tremula x tremuloides*) trees carrying a *GUS* reporter gene under the control of the 1951-bp *PtERF85* promoter (Figure 1d,e). GUS staining confirmed the expression pattern obtained from AspWood, showing expression of *PtERF85* mainly in the phloem and expanding xylem cells, but also in contact ray cells next to vessel elements.

### 3.2. Ectopic PtERF85 Induction Did Not Change Ethylene Response in Hybrid Aspen Wood

The observed induction of *PtERF85* expression by ethylene/ACC [15] prompted us to further investigate the role of *PtERF85* in the ethylene signaling pathway. GUS reporter assays suggested that the stem expression pattern of *PtERF85* was not altered by a 10 h 100 µM ACC treatment and remained concentrated predominantly to the cambium and the xylem (Figure 2a). It should be noted that, compared to *pPtERF85::GUS* expression in greenhouse-grown trees (Figure 1d,e), GUS staining was faint in stems of in vitro grown trees (Figure 2a), making quantification of any response impossible. To study the function of PtERF85 in ethylene-regulated secondary growth, we examined the effect of ACC on transgenic stems of hybrid aspen trees that ectopically express *PtERF85* under the xylem-specific *pLMX5* promoter (“ERF85OE”, described in [13]). According to AspWood (Appendix A), and previous promoter GUS activity studies [13], the expression of *LMX5* was highest during SCW formation. WT trees showed only very low *PtERF85* expression during SCW formation (Figure 1e and Appendix A). Using *pLMX5,* therefore, extended and increased *PtERF85* expression beyond the cambium and xylem cell expansion zone into the SCW formation zone (Appendix A), thus spanning all zones that were affected by ethylene application [13,14]. Upon ACC treatment, WT trees showed enhanced cambial activity, reduced vessel frequency, and the induction of gelatinous-layer (G-layer) in xylem fibers [13,14]. These phenotypes were not observed in ethylene-insensitive trees, which expressed a dominant negative version of the ethylene receptor ETR1 driven by the *pLMX5* promoter [13], indicating that ACC-induced xylem growth and G-layer formation requires functional ethylene signaling. In ERF85OE trees, ACC induced xylem growth and G-layer formation and favored fiber over vessel formation similarly to WT trees (Figure 2b). We detected patchy G-fiber occurrence even in water-treated control trees (indicated by arrowheads in Figure 2b and magnified in Appendix A), which can randomly occur in fast-growing *Populus* trees. We did not confirm a consistent G-fiber induction in ERF85OE trees in additional experiments where we investigated the wood anatomy of greenhouse-grown plants (Figure 3b). ACC treatment also reduced stem height growth of the ERF85OE lines (Figure 2c), another phenotypic response seen in response to ACC application to *Populus* trees [13]. Together with previous results [15], this indicates that, even though *PtERF85* can be transcriptionally induced by ethylene in an ethylene-signaling dependent way, it did not mimic constitutive ethylene signaling when ectopically expressed under *pLMX5* in woody tissues. Thus, ACC treatment was still able to induce previously characterized ethylene responses during wood development.

### 3.3. ERF85OE Lines Showed Increased Xylem Fiber Diameter

In order to understand how the expansion of the expression zone of *PtERF85* under the *pLMX5* promoter would affect xylem formation, we undertook a detailed analysis of the xylem anatomy and subsequently assessed its downstream targets during wood development (Section 3.4). In agreement with previous experiments, two out of three analyzed ERF85OE lines showed a reduced stem growth (Figure 3a; [15]). Observation of ultrathin transverse sections either through transmission electron microscopy or, after toluidine blue staining, through light microscopy, suggested changes in xylem fiber morphology (Figure 3b). Secondary xylem anatomy of the ERF85OE lines was analyzed in xylem areas with clearly visible SCWs, approximately 1 mm radially inwards of the cambium (Appendix A). The average number of fibers (per 149,295 µm^2^ area) was reduced in all transgenic lines, up to 16% (line 5) compared to WT (Figure 3c). No significant differences (*p*-value < 0.05) were observed in the total number of ray or vessel cells between WT and any of the ERF85OE lines (Figure 3d,e, covering an area of 3.02 mm^2^ shown in Appendix A). The fact that the number of fibers in a defined cross-section area of the secondary xylem was reduced in the ERF85OE lines, but the numbers of the ray cells or vessels were not altered, suggests an increase in fiber diameter. Indeed, the fiber diameter, defined in the transverse sections as the area encircled by the middle lamella, increased by up to 27% (line 5) (Figure 3f). No consistent change was observed in vessel diameter (Appendix A). We next analyzed the effect of ectopic *PtERF85* expression in the cell wall and mechanical properties. Using transmission electron microscopy, we observed that fibers had thinner cell walls compared to WT (Figure 3g). Quantification of fiber cell wall thickness (defined as distance between cell-to-cell lumen) on toluidine blue stained ultrathin stem cross-sections showed a reduction (between 10–19%) in cell wall thickness for ERF85OE lines compared to WT (Figure 3g). Furthermore, ERF85OE trees showed a reduction in wood density, ranging from 8 to 12% compared to WT trees (Figure 3h). The decrease in xylem density in ERF85OE lines is therefore likely the result of thinner fiber cell walls and an increased fiber diameter. Taken together, ectopic expression of *PtERF85* throughout all phases of wood development resulted in increased fiber expansion and reduced SCW deposition, and ultimately in reduced wood density.

### 3.4. Ectopic Expression of PtERF85 Activated Expression of Genes Linked to Cell Expansion and Antagonized Induction of Genes Related to SCW Formation

To understand the molecular mechanisms underlying PtERF85 function in xylem differentiation, transcriptome data were generated by RNA-Seq from developing xylem of WT trees and the ERF85OE lines (Figure 4). PCA on normalized gene count data revealed separation of the WT samples from the ERF85OE samples in the first component (explaining 61% of the overall variation), suggesting that genotypic differences explain most of the variation among the transcriptomes (Figure 4a). As indicated by the separation of the three ERF85OE lines according to the second component, slight transcriptome variability was observed also among the three transgenic lines. The expression of *PtERF85* itself, however, was clearly enhanced in all three lines and not significantly different among the three transgenic lines (Appendix A). Differential analysis of genes showing at least a two-fold change between ERF85OE and WT (at a False Discovery Rate (FDR) cutoff of 1%; Appendix A) identified 1893 genes with higher expression in ERF85OE compared to WT, while expression of 410 genes was down-regulated (Figure 4b). Genes with down-regulated expression in ERF85OE were enriched in Gene Ontology (GO) terms related to metabolism and signaling (Appendix A; Appendix A), including homologs with known function in xylem cell wall composition (Figure 4c; e.g., *IRX10 (Irregular X**ylem10), 4CL2 (4-Coumarate:CoA Ligase2*)) and xylem development (e.g., *XCP2 (Xylem Cysteine Peptidase2*), and *CLV20 (CLAVATA20)*). Expression of the *PIN3* homolog was also suppressed in ERF85OE (Figure 4c,d), whose function in polar auxin transport has been linked to increased xylem formation in Arabidopsis [32]. DEGs with up-regulated expression in ERF85OE were enriched (*p*-value < 0.01) in GO terms related to protein translation (n = 335; Appendix A; Appendix A) including genes encoding ribosome subunits, elongation factors, cyclophilins, and chaperones. In addition, we also identified homologs with reported function in xylem development and cell elongation (Figure 4c), including *CEL* encoding an endo-1,4-ß-glucanase that affects xylem development and cell wall thickness in Arabidopsis [33] and cell growth and cellulose content in poplar cell suspension cultures [34]. The observed change in expression was validated for both up- and down-regulated DEGs, using qPCR (Figure 4d). We next assessed the stem expression profile of the up-regulated DEGs in AspWood (Figure 4e). About 67% (1304 out of 1893 genes) of the DEGs were present in AspWood clusters that showed, like *PtERF85*, the highest expression in the cambium and during xylem cell expansion, but low expression during SCW formation (clusters “f”, “e”, “h” in [3]). Approximately 33% of all down-regulated DEGs (137 out of 410 genes) belonged to AspWood clusters dominated by genes with opposite expression profiles to *PtERF85*, meaning highest expression during SCW formation and low expression in cambium and expansion zone (cluster “g” in [3]). Thus, ectopic expression of *PtERF85* might act positively on the expression of gene networks underlying secondary xylem cell expansion, while it has a negative effect on the onset of gene expression in the networks that underly SCW formation.

We utilized the AspWood co-expression network to cluster DEGs according to their expression profile during xylem development (Figure 5). This captured 801 DEGs with high co-expression, which clustered into 13 gene modules (Figure 5a; Appendix A). According to their stem expression profiles and functional characterization, we classified them into gene modules underlying xylem expansion (M1–M5), the transition from primary wall to SCW (M6, M7), SCW formation (M9–M11), and fiber cell death or ray development (M8, M12, M13). Cell expansion-related modules M1–M4 comprised the highest number of DEGs with increased expression in ERF85OE and a *PtERF85*-like stem expression profile (Figure 5a,b). These four expansion-related modules consisted of genes related to protein translation and folding, such as elongation factors (e.g., *Eukaryotic Initiation Factor4A-2 (PtEIF4A2)*, RNA binding (e.g., *RNA-Binding Protein47C (PtRBP47C)*), rRNA processing and maturation (which is partially controlled by Histone deacetylase2C (PtHD2C) [35]), ribosomal subunits (e.g., *Ribosomal Protein S27 (PtRPS27A)*), and chaperones (e.g., *chaperonin-60 Beta2 (PtCPNB2*), Appendix A). Module (M5) contained genes which showed a peak in expression during xylem cell expansion but, in contrast to M1–4, remained highly expressed at the interface between cell expansion and the onset of SCW formation. This module included genes that encode pectin-degrading polygalacturonases (e.g., *PtPG2*, *PtPGX3*) and an ATP-binding cassette transporter. Interestingly, the *PtPGs* that were lowly expressed in the SCW zone in AspWood (e.g., *PtPG9*, *PtPG2*) were up-regulated in ERF85OE, whereas the *PtPGs* that were highly expressed in the SCW zone in AspWood (e.g., *PtPG6*) were repressed in ERF85OE. Modification of the *PtPGs* transcript levels in ERF85OE could lead to loosening of pectin structures to facilitate cell expansion, as for example shown for the *PtPG28/43* homolog in Arabidopsis *PGX3*, which might also influence the pectin composition in SCWs [36]. Also, the two modules M6 and M7 (Figure 5c) contained genes which in AspWood showed high expression during the transition phase from primary cell wall to SCW in AspWood, including genes involved in flowering (e.g., cyclin *DOF2 (PtCDF2)*), response to light (e.g., *Cryptochrome1 (PtCRY1)*) and the circadian clock (e.g., *Reveille8 (PtRVE8)*), as well as the negative regulator of vessel formation *VND-Interacting2 (PtVNI2)*. The three SCW-associated modules (M9–M11, Figure 5D) contained genes associated with auxin signaling (e.g., *PtPIN3, PtIAA4*), biosynthesis of the SCW-specific hemicellulose xylan (e.g., *PtIRX10*) and lignin (e.g., *Pt4CL2*) and xylem maturation (e.g., *PtXCP2*). The expression of all the genes in these three modules was down-regulated in ERF85OE, which agreed with the reduction of SCW formation in the transgenic trees (Figure 3g). Finally, three gene modules (M8, M12, M13, Figure 5e) showed the highest expression in the cell death zone. While expression of the genes belonging to M8 and M12, such as *Chalcone Synthase (PtCHS)*, which is involved in biosynthesis of flavonoids and can prevent auxin transport, was increased in ERF85OE, the expression of genes belonging to M13 (such as *Ammonium Transporter2 (PtAMT2)*) was down-regulated in ERF85OE. In agreement with increased fiber diameters in ERF85OE lines, we observed that ectopic expression of *PtERF85* throughout all stages of xylem development induced genes specific to xylem cell expansion. At the same time, genes with the highest expression during SCW formation were down-regulated in ERF85OE. Thus, radially prolonged transcriptional activation of genes involved in xylem cell expansion, and the consequent indirect suppression of SCW-related gene expression could ultimately cause the observed reduction of fiber cell wall thickness.

### 3.5. PtERF85-Mediated Transcriptional Regulation Mechanisms during Xylem Cell Expansion

We next compared Arabidopsis homologs identified as DEGs in our dataset with direct targets of AtCRF4 in Arabidopsis [37,38]. Of the 2303 DEGs, 2227 had corresponding Arabidopsis homologs. However, because of gene duplication events in *Populus*, 475 DEGs did not have unique Arabidopsis homologs. In total, we identified 1752 unique homologs among the initial 2303 DEGs. Dataset comparison to the targets of AtCRF4 revealed that 995 out of these 1752 Arabidopsis homologs had been identified as direct AtCRF4 targets (Appendix A). Their *Populus* homologs are therefore potential direct targets of *PtERF85*. This included both homologs of up- and down-regulated DEGs in ERF85OE (Figure 6a). These results point towards a potential direct regulation of DEGs by PtERF85. In support of this, promoter regions (from 500 up to 2000 bp upstream of the start codon) of up-regulated DEGs showed significant enrichment of the ERF-binding motif GCC-Box (*p*-value < 6.01 × 10^−11^ and *p*-value < 1.87 × 10^−5^, for 0.5 and 2 kbp, respectively). However, no statistically significant enrichment was identified for this motif in promoters of down-regulated DEGs (Appendix A). To determine how PtERF85 could achieve transcriptional regulation during different developmental stages in xylem tissues, we analyzed co-expression relationships of the potential PtERF85-regulated TFs during wood development (Figure 6b, Appendix A). Potential PtERF85-regulated *TFs* were grouped based on their expression profile in AspWood into *TFs* with low expression during SCW formation (e.g., *PtGATA15, PtHD2C*; Figure 6b) and TFs with high expression during SCW formation (e.g., *Heat Shock transcription Factor3 (PtHSFB3), PtWRKY21, PtERF151*). The GCC-box (AGCCGCC) and the DRE-motif (GTCGGT/C) were present in promoters (2 kb) of *TFs* from both groups (Appendix A), suggesting a potential direct transcriptional regulation of certain targets by PtERF85. Indeed, the closest homolog of *PtERF85* in Arabidopsis, *AtCRF4*, controlled expression of 18 *TF* homologs (out of all the 57 *TFs* identified in our study, *p*-value = 0.17) by direct binding according to data obtained in Varala et al. (2018) ([37]; highlighted in Figure 6b, Appendix A). Among them, we identified homologs of *PtHD2A* and *PtGATA17L*. We ranked each *TF* based on its network degree (number of neighbors) with the aim to select *TFs* with a potential central regulatory function downstream of PtERF85. Several *PtGATAs* (*PtGATA15, PtGATA16,* and *PtGATA17L*) and two histone deacetylases (*PtHD2A* and *PtHD2C*), which we included because they regulate gene expression by promoter binding [35,39,40], showed a central position in the *TF* network downstream of PtERF85. Their expression during wood formation followed the same pattern as *PtERF85* and although only *PtHD2A* and *PtGATA17L* were identified as direct target genes of AtCRF4 [37], we identified the GCC-box (*PtGATA17L*) or DREB-motif (*PtGATA15, PtHD2C*) in their promoter region (Appendix A). *PtHD2A, PtHD2C,* and *PtGATA17L* have been identified in gene modules up-regulated during cell expansion (Figure 5b) and homologs of DEGs from ERF85OE were found as direct target genes of AtGATA17L and AtHD2C in Arabidopsis (Figure 6a, Appendix A; [35,38]). Furthermore, according to the AspWood co-expression network analysis (Figure 6b; Appendix A), expression of *PtGATAs* (like *PtGATA15* in gene module M4) and both, *PtHD2C* and *PtHD2A* (present in gene module M2), was linked to the expression of genes encoding ribosomal subunits, chaperones, and elongation factors. These results point towards a central role of PtGATA17L and PtHD2C in transcriptional regulation of ribosome biogenesis and translation downstream of *PtERF85* during xylem cell expansion. Concurrently, ectopic expression of PtERF85 could either directly through PtERF85 itself or other *TFs* prevent the onset of SCW formation by suppressing central TFs in the underlying network, such as *PtHSFB3* or *PtERF151* (both containing a DRE-motif in their promoter and belonging to gene module M10; Appendix A).

## 4. Discussion

### 4.1. PtERF85 Modulates the Balance between Xylem Cell Expansion and SCW Formation through Transcriptional Regulation

In situ and in silico expression analyses detected a peak of *PtERF85* expression in the phloem, cambial, and expanding xylem (Figure 1). This agrees with the expression pattern observed for the closest homolog of *PtERF85* in Arabidopsis, *AtCRF4*, which is also highly expressed in phloem and xylem tissues [22]. Ectopic expression of *PtERF85* throughout xylem cell expansion and SCW formation caused an increase in xylem fiber diameter, while simultaneously reducing SCW thickness and wood density in hybrid aspen (Figure 2). Interestingly, although the transcriptional induction of *PtERF85* by ACC suggests that its function is regulated by ethylene, the xylem cell phenotypes we observed in the ERF85OE lines do not represent phenotypes typically observed in the wood of ethylene/ACC-treated stems [13,14] or ectopic expression of other ethylene-regulated *PtERFs* [15,18] in hybrid aspen. Upon treatment with ACC, ERF85OE lines showed enhanced cambial activity and G-layer formation as previously observed in wildtype trees [14], indicating that ERF85OE lines respond normally to ACC. A previous transcriptome study revealed that *PtERF85* is induced by a 10 h ACC application downstream of the ethylene signaling pathway [14]. We could not confirm this ACC-responsiveness using *pPtERF85::GUS* lines in our experiments. It should however be noted that the GUS expression was rather weak under in vitro conditions (used for ACC treatments) as compared with greenhouse-grown trees (Figure 1 and Figure 2).

The differences in wood morphology and density in ERF85OE correlated with an induction of expansion zone-associated genes and repression of SCW-associated genes (Figure 4e) as well as with de-regulation of transcriptional networks in xylem cell expansion and SCW formation (Figure 5). Genes that affect cell elongation, like *CEL1* [33,34] and *GAST1* [41], were up-regulated by ectopic expression of *PtERF85* (Figure 4c). In contrast, expression of genes involved in xylem maturation (e.g., *VNI2, XCP2*), lignin biosynthesis (e.g., *4CL2, CHS*), and SCW deposition (e.g., *IRX10, GH9C2, GH9B5*) were downregulated in ERF85OE (Figure 4c and Figure 5d). Two possible scenarios can explain this finding. Either PtERF85 has a dual function and, at once, induces genes related to the cell expansion zone, while directly repressing those specific to the SCW formation zone, or it only induces genes related to the cell-expansion zone and it is the prolongated expression of these expansion-associated genes that indirectly then delays the onset of SCW gene expression in a spatio-temporal manner. Alternatively, a combination of both mechanisms may be at the origin of the observed phenotypes of enhanced fiber expansion and reduced SCW thickness in ERF85OE lines. It is widely accepted that TFs can function as repressors or activators depending on their target and interacting factors. Out of 1752 unique Arabidopsis genes found among all homologs of PtERF85-regulated DEGs, 995 have also been identified as direct targets of AtCRF4 in Arabidopsis (Figure 6a; Appendix A) and those contain both up- and down-regulated genes in our dataset. This favors the assumption that PtERF85 could indeed have a dual regulatory function depending on its targets. On the other hand, induction of *CEL1* (Figure 4c) might trigger cell growth, while suppressing SCW thickening as seen in Arabidopsis [33]. It is likely that the phenotype is caused by a combined direct and indirect action of PtERF85.

### 4.2. PtERF85 Might Control Xylem Cell Wall Properties to Allow Cell Expansion

In line with prolonged cell expansion and increased fiber diameter in ERF85OE, we also observed transcriptional changes in genes regulating cell wall properties, particularly pectin, required for the regulation of cell expansion. Genes encoding enzymes with potential homogalacturonan backbone degradation function (*PtPG55*, *PtPG2*, *PtPG9*) were up-regulated, indicative of increased pectin loosening that facilitates cell expansion, as for example shown for the *PtPG28/43* homolog in Arabidopsis *AtPGX3* [42]).

Modification of poplar *CEL1,* an endo-1,4-β-glucanase expression has been shown to correlate with cell growth and cellulose content in suspension-cultured poplar [34]. These phenotypic responses were pronounced in the presence of auxin and sucrose [34]. Indeed, auxin is known to stimulate cell expansion by inducing cell wall loosening [43]. Hereby, auxin triggers demethylesterification of pectins, which can lead to enhanced wall extensibility in the presence of pectin degrading enzymes, like polygalacturonases [44]. To gain cell wall rigidity in SCWs, auxin must be exported by auxin transporters like PIN3, which has been suggested to transport auxin in *Populus* stems [45]. In agreement with this fact, expression of *PtPIN3* is up-regulated during SCW formation in the Aspwood data (representing normal wood growth in WT trees; Figure 5d). Expression of *PtPIN3* and auxin signaling genes (e.g., *PtIAA4, PtIAA11,* and *SAUR-*like genes) was down-regulated in transgenic ERF85OE trees, suggesting that PtERF85 modifies auxin fluxes. The potential connection between PtERF85 and polar auxin transport via PIN proteins agrees with previous reports that showed direct regulation of *AtPIN1* and *AtPIN7* by *AtCRF2* and *AtCRF6* in Arabidopsis [31]. Thus, PtERF85 could enhance cell expansion by transcriptional regulation of genes encoding polygalacturonases, CEL1, and proteins involved in auxin transport.

### 4.3. PtERF85-Mediated Regulation of Translation could Affect the Dynamics of Cell Growth in Expanding Xylem

Ectopic expression of *PtERF85* induced the expression of the molecular machinery involved in protein synthesis (Appendix A; Appendix A), including genes encoding ribosomal subunits, RNA polymerases, and modulators of rRNAs and pre-RNAs.

A recent study in Arabidopsis revealed a new role of histone deacetylases (AtHD2B and AtHD2C) in transcriptional and post-transcriptional regulation of ribosome biogenesis [35]. Expression of two histone deacetylases, *PtHD2A* and *PtHD2C,* was strongly up-regulated in our ERF85OE lines (Figure 6b, Appendix A). Furthermore, gene regulatory network analysis also links expression of *PtERF85*, *PtHD2C,* and *PtGATAs* (*PtGATA15* and *PtGATA17L*; Figure 6a). In agreement with this, we identified the DRE-motif in the promoter of *PtGATA15* and *PtHD2C,* and a GCC-box in the promoter of *PtGATA17L*. Furthermore, *AtGATA17L* was identified as a direct target of AtCRF4 in Arabidopsis (Figure 6a; Appendix A; [37,38]), suggesting that ERF-mediated transcriptional regulation of the gene network underlying xylem cell expansion might involve PtGATA15, PtGATA17L, and PtHD2C as co-factors or potential second tier transcription factors. Future work will be required to identify direct targets of PtERF85 and the molecular and functional importance of PtERF85-regulated *TFs*.

## 5. Conclusions

Our study revealed PtERF85 as a regulator of transcriptional programs underlying xylem cell expansion and SCW formation. PtERF85 stimulates fiber cell expansion and represses SCW formation probably through a combination of direct and indirect mechanisms. We, therefore, suggest that PtERF85 acts as a control point for the regulation of transcriptional programs during these two developmental phases of secondary xylem development.

## Figures and Tables

**Figure 1 cells-10-01971-f001:**
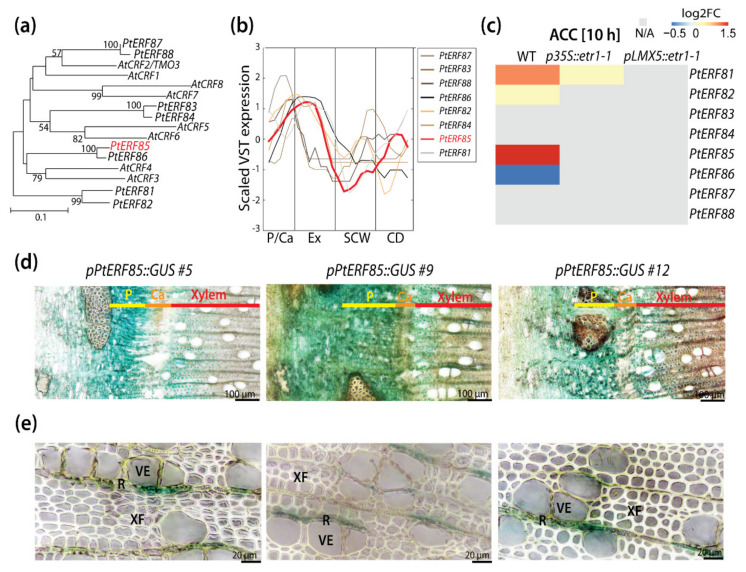
Expression of *PtERF85* was observed during cambial growth and the expansion of xylem cells. (**a**) Phylogenetic tree of PtCRFs (cluster VI; PtERF81-PtERF88) and their homologs in *A. thaliana* (At) (modified from [15]). (**b**) Expression profile of *PtCRFs* in aspen stems. Data (shown for one representative tree (T1)) was obtained from the transcriptome atlas AspWood [3]. To emphasize expression differences, scaled and smoothened VST expression values (implemented function in http://popgenie.org, last accessed on 19 November 2019) are shown. (**c**) Heatmap showing the expression pattern of *PtCRFs* (log2 fold changes) in xylem tissue of WT and two ethylene-insensitive trees (*p35S::etr1-1*, *pLMX5::etr1-1*) in response to 10 h-treatment with 10 µM ACC (data extracted from [14]). (**d**,**e**) Cross sections of greenhouse grown trees showing GUS activity in three transgenic hybrid aspen trees expressing *pPtERF85::GUS* reporter gene. P = phloem; Ca = cambium; Ex = expanding xylem; SCW = secondary cell wall formation; XF = xylem fiber; VE = vessel element; R = ray. Gene names refer to *PtERF81 (Potri.019G131300), PtERF82 (Potri.013G158500), PtERF83 (Potri.002G167400), PtERF84 (Potri.014G094500), PtERF85 (Potri.015G023200), PtERF86 (Potri.012G032900), PtERF87 (Potri.001G094800)*, and *PtERF88 (Potri.003G136300)*.

**Figure 2 cells-10-01971-f002:**
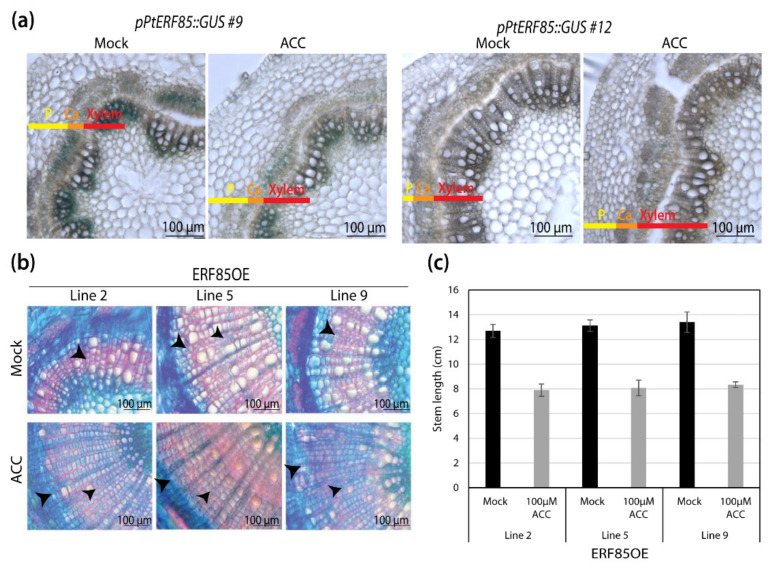
ERF85OE lines showed enhanced ethylene-mediated stem phenotypes. (**a**) GUS staining in two transgenic lines that carry a *pPtERF85::GUS* reporter. Transgenic reporter lines were treated for 10 h with either water (mock) or 100 µM ACC. (**b**) Stem cross-sections from WT, transgenic ethylene-insensitive trees (*pLMX5::etr1-1*; [13]), and ERF85OE lines. Trees were treated with 10 µM ACC for 14 days. Cross-sections (50 µM-thick) were stained with safranine and toluidine blue and magnified using a 20-fold objective. Arrows highlight the onset of a gelatinous (G)-layer in xylem fibers [14]. (**c**) Plant height of ERF85OE lines in response to water (mock) or 100 µM ACC treatment for 14 days. Bars represent the average height (+/− standard error) obtained from five biological replicates.

**Figure 3 cells-10-01971-f003:**
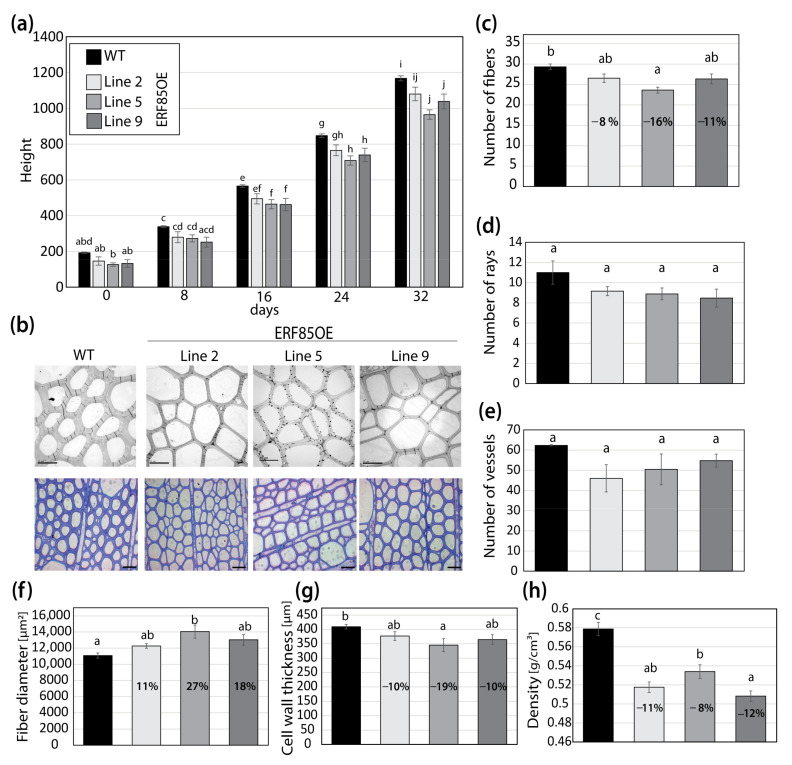
Ectopic expression of *PtERF85* in woody tissue increased fibers diameter and reduced cell wall thickness. (**a**) Stem growth rate of WT and three transgenic lines that ectopically express *PtERF85* (ERF85OE) under a xylem specific wood promoter (*pLMX5*; [15]) in a controlled greenhouse environment. (**b**) Transmission electron micrographs (TEM; 100-fold magnification; scale bar is 10 µm) and toluidine blue staining (20-fold magnification; scale bar is 40 µm). TEM pictures cover a total area of 149,295 µm^2^. (**c**) The number of fiber cells counted in toluidine blue cross-sections is shown in (**b**). (**d**,**e**) Numbers of ray and vessel cells (in an area of 3.02 mm^2^) in WT and ERF85OE lines. Cross-sections used for quantification are shown in Appendix A. (**f**) Fiber diameter from toluidine blue-stained cross-sections is shown in (**b**). Cell outlines were defined by the middle lamella and the enclosed area [µm^2^] was measured using ImageJ. (**g**) Cell wall thickness (determined as the lumen-to-lumen distance of two neighboring cells [µm]) based on toluidine blue-stained stem sections (shown in (**b**)), using a 100-fold magnification. (**h**) Wood density. For all panels, bars represent mean ± SE calculated from three biological replicates per line using a linear effect model. Significant differences between genotypes were indicated by the unique occurrence of letters above the bars and were assigned based on multiple comparison tests and a *p*-value < 0.05.

**Figure 4 cells-10-01971-f004:**
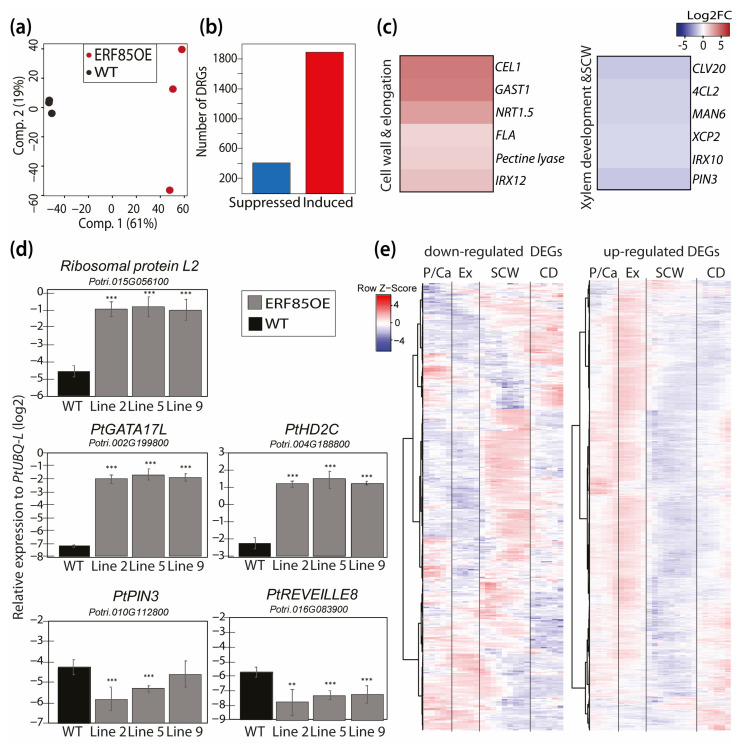
Ectopic expression of PtERF85 stimulated the expression of genes linked to cell expansion and suppressed genes linked to SCW formation during xylem differentiation. (**a**) PCA of xylem transcriptome data obtained from three WT samples (WT, each sample represents a pool of three plants) and three ERF85OE lines (2,5,9). (**b**) Distribution of (DEGs in ERF85OE. DEGs were selected based on a |log2FC| > 1 and pAdj < 0.01 compared to WT. (**c**) Representative genes found among the DEGs in ERF85OE lines with known function of their homologs in Arabidopsis related to cell elongation, cell wall composition, and xylem development (*Potri.015G127900 (CEL1); Potri.002G022600 (GAST1); Potri.001G171900 (Pectate lyase); Potri.008G064000 (IRX12); Potri.014G156600 (CLV20); Potri.010G112800 (PIN3); Potri.005G256000 (XCP2)) and SCW composition (Potri.010G192300 (FLA); Potri.001G036900 (4CL2); Potri.016G138600 (MAN6); Potri.005G256000 (IRX10)), Potri.003G088800 (NRT1.5*). (**d**) Validation of differential expression of up- and down-regulated DEGs using qPCR. Bars represent the mean expression level+- SD of target genes relative to the housekeeping gene *PtUBQ-L* (log2). Asterisks indicate statistically significant differences between WT and ERF85OE lines according to the Student’s *t*-test (with *p*-value < 0.01 (**), and *p*-value < 0.001 (***)). (**e**) Heatmap showing the expression profile of up- and down-regulated DEGs during stem development in hybrid aspen. Data were obtained from the AspWood database [3]. Genes were scaled so that red represents the highest expression level for a given gene, and blue the lowest, across all developmental zones. P/Ca = phloem/cambium, Ex = expanding xylem, SCW = secondary cell wall formation, CD = cell death.

**Figure 5 cells-10-01971-f005:**
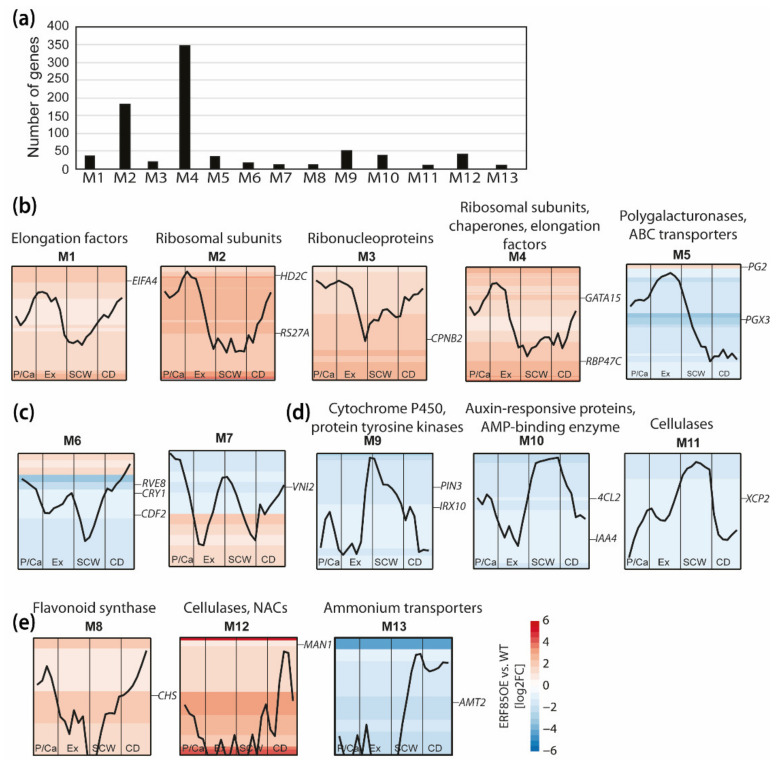
Ectopic expression of *PtERF85* induced expression of genes related to translation, while suppressing expression of SCW-related genes. (**a**) Genes with similar expression profiles during stem development were categorized into 13 gene modules (M1–M13) based on the AspWood co-expression network [3]. The size of each gene module is given in the histogram. (**b**) Gene modules with highest expression during xylem cell expansion. A representative gene, defined as the gene with the highest level of expression, was chosen to illustrate the expression profile of genes in each module (Appendix A). Enriched PFAM terms (*p*-value < 0.05) are listed above each module (Appendix A). Background heatmaps indicate gene expression (log2FC) in ERF85OE compared to WT for the genes in the module (each row represents a gene). (**c**) Gene modules with the highest expression levels in the transition phase of primary cell wall to SCW formation. (**d**) Gene modules with the highest expression level during SCW formation. (**e**) Gene modules with the highest expression level in the zone of cell death. P/Ca = phloem/cambium; Ex = expanding xylem; SCW = secondary cell wall formation, CD = cell death; *EIFA4-2 (Potri.001G197900); HD2C (Potri.004G188800); RS27A (Potri.003G161200); CPNB2 (Potri.001G002500); GATA15 (Potri.008G213900); RBP47C (Potri.002G128100); PG2 (Potri.001G171900); PGX3 (Potri.008G100500); RVE8 (Potri.016G083900); CDF2 (Potri.004G121800); CRY1 (Potri.005G164700); VNI2 (Potri.017G063300); CHS (Potri.003G176700); PIN3 (Potri.010G112800); IRX10 (Potri.003G162000); 4CL2 (Potri.001G036900); IAA4 (Potri.010G078400); XCP2 (Potri.005G256000); MAN1 (Potri.001G266900); LBD4 (Potri.007G066700); AMT2 (Potri.019G000800)*.

**Figure 6 cells-10-01971-f006:**
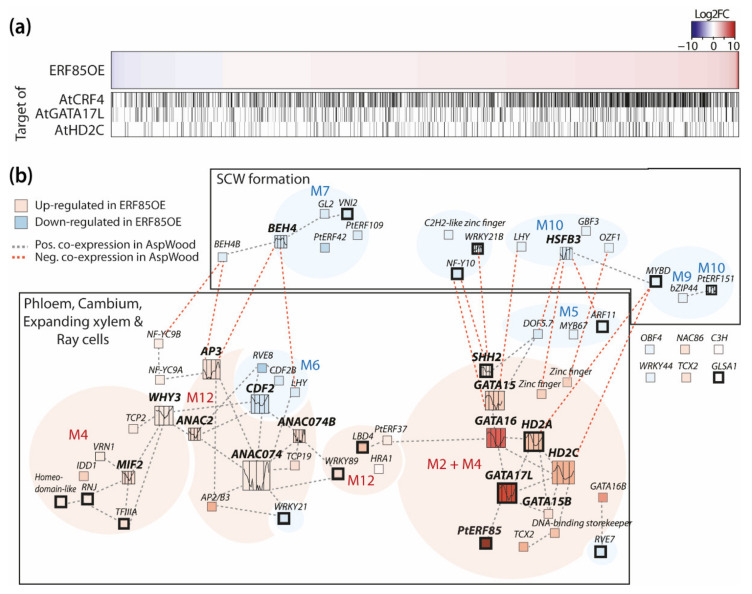
Potential PtERF85-controlled gene regulatory networks during wood development. (**a**) Heatmap showing the gene expression of all DEGs in ERF85OE compared to WT. The graphs beneath the heatmap indicate homologs that were identified as direct targets of AtCRF4, AtGATA17L, and AtHD2C in Arabidopsis (Appendix A; [35,37,38]). (**b**) Co-expression network of differentially expressed TFs in ERF85OE during wood development (Appendix A). Thickened box lines indicate TFs for which homologs were identified as direct targets of AtCRF4 in Arabidopsis [37]. Background color represents gene expression induction (red) or suppression (blue) in ERF85OE lines compared to WT. TFs with at least four neighbors were highlighted by increased box size and expression profiles inside the box show their expression pattern during wood formation (obtained from the AspWood database [3]). Grey lines indicate positive co-expression between the TFs, red indicates negative co-expression. M = module (corresponding to Figure 5).

## Data Availability

Raw files can be downloaded from the European Nucleotide Archive (ENA) under PRJEB35743. Data quality summary and scripts used in this study were deposited to https://github.com/carSeyff/PtERF85OE (accessed on 18 May 2021).

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
