# Peer review of "PopulusPtERF85 Balances Xylem Cell Expansion and Secondary Cell Wall Formation in Hybrid Aspen"

_cells, 2021, doi:10.3390/cells10081971_

Round 1

Reviewer 1 Report

The manuscript “Populus ERF85 balances xylem cell expansion and secondary cell wall formation in hybrid aspen” is quite interesting paper to describe the transcription factor ERF85 functions as a switch between different phases of xylem differentiation during wood formation. all the figures and results are clear and well done. So I prefer to accept after minor revision.

Minor comments

1. In line 336, in the sentence "the result of thinner fiber walls and an increased fiber diameter", "fiber walls" is confusing, it may be "cell wall" or "fiber cell wall".

2. In line 504, there is an error, according to your description "Indeed, the closest homolog of ERF85 in A. thaliana, AtCRF4, controlled expression of 18 TF homologs (out of all 57 TFs identified in our study, p-value=0.17) by direct binding ( highlighted in Figure 5(b), Supplement Table S2; [38]).", this should be "highlighted in Figure 6(b)".

3. In line 595-598, the previous sentence says "expression of PtPIN3 is up-regulated", but the latter sentence says "Expression of PtPIN3 and auxin signaling genes (e.g. PtIAA4, PtIAA11, SAUR-like genes) was down-regulated", which seems a bit contradictory.

Author Response

  1. In line 336, in the sentence "the result of thinner fiber walls and an increased fiber diameter", "fiber walls" is confusing, it may be "cell wall" or "fiber cell wall".

We thank Reviewer 1 for highlighting these three points. All raised points have been modified in the manuscript accordingly (marked in yellow).

“The decrease in xylem density in ERF85OE lines is therefore likely the result of thinner fiber cell walls and an increased fiber diameter.”

  1. In line 504, there is an error, according to your description "Indeed, the closest homolog of ERF85 in A. thaliana, AtCRF4, controlled expression of 18 TF homologs (out of all 57 TFs identified in our study, p-value=0.17) by direct binding ( highlighted in Figure 5(b), Supplement Table S2; [38]).", this should be "highlighted in Figure 6(b)".

“Indeed, the closest homolog of ERF85 in A. thaliana, AtCRF4, controlled expression of 18 TF homologs (out of all 57 TFs identified in our study, p-value=0.17) by direct binding (highlighted in Figure 6(b), Supplement Table S2; [38]).“

  1. In line 595-598, the previous sentence says "expression of PtPIN3 is up-regulated", but the latter sentence says "Expression of PtPIN3 and auxin signaling genes (e.g. PtIAA4, PtIAA11, SAUR-like genes) was down-regulated", which seems a bit contradictory.

To gain cell wall rigidity in SCWs, auxin must be exported by auxin transporters like PIN3, which has been suggested to transport auxin in Populus stems [45]. In agreement with this fact, expression of PtPIN3 is up-regulated during SCW formation in the Aspwood data (representing normal wood growth in wild-type trees; Figure 5(d)). Expression of PtPIN3 and auxin signalling genes (e.g. PtIAA4, PtIAA11, SAUR-like genes) was down-regulated in transgenic ERF85OE trees, suggesting that ERF85 modifies auxin fluxes.

We attach for reference the changed manuscript with highlighted edits.

Reviewer 2 Report

The article is a well developed full study on the influence of ERF85 on secondary cell wall formation in aspen. I have these comments which might help improve the article.

Question about Figure 2c: The ethylene-insensitive tree seems to be responding to ethylene by showing a decrease in stem length. Was this expected? The article is focusing on the overexpression line, which is not affected compared to wild type, but the whole set of results is possibly not really looking at an ethylene response?

Question about RNA-Seq data: There is variation in the transcriptomes of the overexpression lines based on the PCA analysis. Is it possible that in some cases, the ERF85 gene is actually being suppressed in some tissues in the overexpression lines, thus causing a loss of function effect in addition to a gain of function effect? It would help the analysis to see expression data of ERF85 in the different lines and explore and report the variation in the three lines. It is not totally clear how the data from the three lines was combined to compare to wild-type or were individual comparisons done with each overexpression line?

Final question: Is there any speculation about what might be down-regulating ERF85 as xylem cell differentiation occurs to allow for secondary cell wall formation?

Author Response

Question about Figure 2c: The ethylene-insensitive tree seems to be responding to ethylene by showing a decrease in stem length. Was this expected? The article is focusing on the overexpression line, which is not affected compared to wild type, but the whole set of results is possibly not really looking at an ethylene response?I

We would like to thank Reviewer 2 for raising this question. We have not included the loss of stem height or xylem growth induction upon ACC treatment for ethylene-insensitive trees (=pLMX5::etr1-1), because those results have already been shown in a previous publication that introduced those transgenic trees (see Love et al., 2009). We realized that the Figure capture might have suggested that we included the height growth for WT and the ethylene-insensitive trees, but we have now removed this part in the current manuscript (“(c) Plant height of ERF85OE lines in response to water (mock) or 100µm ACC treatment for 14 days.”). While ACC triggers plant height in wild-type trees, this response is lost in the ethylene-insensitive trees and the ERF85OE lines show a growth suppression in response to ACC (mentioned in text in line 298: “ACC treatment also reduced stem height growth of ERF85OE lines (Figure 2(c)), another phenotypic response seen in response to ACC application to Populus trees [13].”).

Question about RNA-Seq data: There is variation in the transcriptomes of the overexpression lines based on the PCA analysis. Is it possible that in some cases, the ERF85 gene is actually being suppressed in some tissues in the overexpression lines, thus causing a loss of function effect in addition to a gain of function effect? It would help the analysis to see expression data of ERF85 in the different lines and explore and report the variation in the three lines. It is not totally clear how the data from the three lines was combined to compare to wild-type or were individual comparisons done with each overexpression line?

We agree with Reviewer 2´s observation that the second principal component indicates transcriptome variation between the ERF85OE lines. However, the GUS staining in Figure 1 indicates that endogenous expression of ERF85 occurs in all zones used to generate this RNA-Seq data set. This also holds true for the expression pattern of LMX5 (see Love et al. 2009). It is not possible to recreate the same experimental set-up used for the Aspwod Data from the transgenic ERF85OE trees, because this can only done from several years old trees with a large stem diameter.

We would like to thank Reviewer 2 for the suggestion of bringing more clarity on the variation of PtERF85 expression in the ERF85OE lines by adding the specific RNA-Seq expression values. We have included the expression levels of PtERF85 in all three WT and ERF85OE lines now as Supplement Figure 4 and added the following text to section 3.4:

The second component separates mainly the ERF85OE lines, explaining approximately 20% of the overall variation, suggesting slight differences in the transcriptomes among those replicates. Expression of ERF85 itself was clearly enhanced in all three analyzed lines and not significantly different among the three transgenic lines (Supplement Figure S4).  

We agree with Reviewer 2´s suggestion regarding the specification of the sample combination for the RNA-Seq analysis and therefore included the following addition in the material and methods part under 2.8:

To identify ERF85-mediated transcriptional changes downstream of ERF85 and independent of line-to-line variation, data from all three ERF85OE lines were combined for further analysis. Similarly, we also combined all three WT replicates.

Final question: Is there any speculation about what might be down-regulating ERF85 as xylem cell differentiation occurs to allow for secondary cell wall formation?

We have not included any speculations in this direction in the current manuscript, but based on reports of the closest homolog of ERF85 in Arabidopsis, AtCRF4, nitrogen/ammonium or temperature could be potential upstream signaling mechanisms that influence ERF85 expression during wood formation. The exact molecular mechanism or the responsible components are not known.

We attach for reference the changed manuscript with highlighted edits.

Reviewer 3 Report

In this paper, a ERF transcription factor is reported which functioned in balancing xylem cell expansion and secondary cell wall formation in hybrid aspen (Populus tremula x tremuloides). Expression of ERF85 can induce the expression of genes involved in cell expansion, translation and growth, and decreased the expression of genes associated with plant vascular development and biosynthesis of SCW chemical components. It is interesting in the analysis of ERF85-controlled gene regulatory networks during wood development combined with a large-scale xylem transcriptome analysis. It is of great significance to studying the regulation of wood formation. I still have some questions.

  1. Based on the available results, “ERF85 functions as a switch between different phases of xylem differentiation during wood formation.”in abstract. this might be a bit over interpreted.
  2. Why were the Growth conditions for trees used for wood density and anatomy analysis different?
  3. In “2.2 Cloning and generation of transgenic trees”, Was only the promoter region of ERF85 inserted into the fusion expression vector? Did the author use the transgenic trees to perform wood density and anatomy analysis without the overexpression of the ERF85 genes? Only the promoter? OK, in results , we can see “hybrid aspen trees that ectopically express ERF85 under the xylem-specific pLMX5 promoter”,so it needs to be supplemented in M.M.
  4. Why did not the author test the chemical components of the transgenetic tree?
  5. No EXP, XET/XHT, PE, PME and so on were identified in the transcriptome? Base on other transcriptome, like ref[3], these genes might be screened, qRT-PCR analysis of these genes might be interesting.
  6. In Fig 2, I can not see the G-layer clearly.Fig 2 C, H2O, 2 should be subscript. Fig 2 B, where are the WT and transgenic ethylene insensitive trees? Fig 2 C, Is the plant shorter because of fewer cells or smaller cells?

Author Response

We would like to thank Reviewer 3 for the interesting suggestions.

  1. Based on the available results, “ERF85 functions as a switch between different phases of xylem differentiation during wood formation.”in abstract. this might be a bit over interpreted.

We have modified this into “ERF85 contributes to the transition of fiber cells from elongation to secondary cell wall deposition.

2. Why were the Growth conditions for trees used for wood density and anatomy analysis different?

The experiments were performed at different times in different laboratories (Helsinki and Umeå university), but with the same transgenic lines. Growth parameters (e.g. stem height) were measured in both experimental set-up to ensure the robustness of the phenotypes seen in the ERF85OE lines from experiment to experiment. Using these independent repetitions thus helped to validate phenotypes.

We have added to the material and methods : Growth conditions for trees used for wood density and anatomy analysis were slightly different, as they were performed in a different lab (Helsinki university) yet similar height growth phenotypes were observed as in the other experiments

3. In “2.2 Cloning and generation of transgenic trees”, Was only the promoter region of ERF85 inserted into the fusion expression vector? Did the author use the transgenic trees to perform wood density and anatomy analysis without the overexpression of the ERF85 genes? Only the promoter? OK, in results , we can see “hybrid aspen trees that ectopically express ERF85 under the xylem-specific pLMX5 promoter”,so it needs to be supplemented in M.M.

As the ERF85OE lines (under pLMX5) have been previously generated and published, we’ve added this information more specifically to the introduction to avoid confusion and also to the material and methods part.

Added to introduction:

Overexpression of ERF85 under the wood specific (pLMX5) promoter reduced hybrid aspen stem diameter and height growth and altered the chemical wood composition, as identified in a large-scale screening for ERFs with an impact on wood formation [15]. In this present article, we explore the phenotype caused by ERF85 overexpression further to understand the function of ERF85 in xylem formation.

Added to material and methods:

pLMX5::ERF85 overexpressor constructs and plant transformation have been previously described [15].

4. Why did not the author test the chemical components of the transgenetic tree?

The chemical composition of the ERF85OE lines has already been described in Vahala et al., 2013, we mention this now in the introduction (see answer to point 3).

5. No EXP, XET/XHT, PE, PME and so on were identified in the transcriptome? Base on other transcriptome, like ref[3], these genes might be screened, qRT-PCR analysis of these genes might be interesting.

Indeed, we have extensively studied typical primary and secondary cell wall marker genes within the pool of DRGs in ERF85OE. In case we found an interesting candidate, we have highlighted it in the text and Figures and relevant candidates were validated by qPCR (Fig 4d). 

6. In Fig 2, I can not see the G-layer clearly.Fig 2 C, H2O, 2 should be subscript. Fig 2 B, where are the WT and transgenic ethylene insensitive trees? Fig 2 C, Is the plant shorter because of fewer cells or smaller cells?

H20 has been changed to “Mock”.

We added a zoomed in version of the microscopy images from Figure 2 as supplement Figure S2 to make the G-layers better visible.

We have not included the ACC-induced changes in WT and transgenic ethylene insensitive trees as this has been reported in detail in Love et al. 2009 (mentioned in text in line 298: “ACC treatment also reduced stem height growth of ERF85OE lines (Figure 2(c)), another phenotypic response seen in response to ACC application to Populus trees [13].”)

The reduced height growth of Populus stems in response to the ethylene precursor ACC is a result of shorter internodes and reduced vessel and fiber length, as analyzed in Love et al. 2009. The reason for the reduced fiber length is likely the result of stimulated cambial cell division, which is observed in response to ACC.

We attach for reference the changed manuscript with highlighted edits.

Reviewer 4 Report

The manuscript by Seyfferth et al., gives an interesting outlook how transcription factor ERF85 play master role in xylem cell expansion and secondary wall formation in woody plant Populus hybrid.

Ethylene influences the secondary cell wall formation along with other environmental factors as evident in hybrid aspen trees. Authors developed the investigation largely around ERF85 identification from TF network data (AspWood database) and mRNA induction on exogenous application as published by their group. The expression pattern found in-silico was validated using GUS expression. Moreover, the visual quantification of stem phenotypes on Ethylene treatment co-relates with negative regulation of secondary cell wall and wood density by ectopic expression of ERF85. Authors are able to pin point definitive downstream targets of transcription factor.The transcriptome analysis and its correlation with Arabidopsis network is very interesting and brought new perspective. The transcriptome data been made publicly accessible is another good part of this work.

This work is promising lead in vascular development biology of woody plant, while also propose new directions to follow.

Author Response

We thank Reviewer 4 for the kind words. No further comments needed to be addressed.